# Multi-View Gait Recognition Based on a Siamese Vision Transformer

Yanchen Yang [1] , Lijun Yun [1,2,*], Ruoyu Li [1], Feiyan Cheng [1] and Kun Wang [1]

1   College of Information, Yunnan Normal University, Kunming 650000, China
2   Yunnan Key Laboratory of Optoelectronic Information Technology, Kunming 650000, China
*   Correspondence: yunlijun@ynnu.edu.cn

**Abstract:** Although the vision transformer has been used in gait recognition, its application in multi-view gait recognition remains limited. Different views significantly affect the accuracy with which the characteristics of gait contour are extracted and identified. To address this issue, this paper proposes a Siamese mobile vision transformer (SMViT). This model not only focuses on the local characteristics of the human gait space, but also considers the characteristics of long-distance attention associations, which can extract multi-dimensional step status characteristics. In addition, it describes how different perspectives affect the gait characteristics and generates reliable features of perspective–relationship factors. The average recognition rate of SMViT for the CASIA B dataset reached 96.4%. The experimental results show that SMViT can attain a state-of-the-art performance when compared to advanced step-recognition models, such as GaitGAN, Multi_view GAN and Posegait.

**Keywords:** multi-view gait recognition; Siamese neural network; vision transformer; view-feature conversion; gradual view

## 1. Introduction

The identification of human individuals based on their gait, alongside a range of biological features, including facial features, fingerprints, and irises, has the benefits of being a long-range, non-intrusive, and passive mode of identification [1]. In addition, as the security facilities of urban and public places are gradually improved, monitoring facilities, such as cameras are ubiquitous, facilitating the use of basic, low-resolution instruments of which the identified target is unaware [2]. Personality traits determine one's identity. This has led to the widespread use of deep-learning-based gait-recognition technology in modern society [3], particularly in criminal investigations and public security; this technique has significant potential for future applications [4]. To sum up, the fact that gait recognition allows for the undetectable identification of individuals means that it has obvious advantages in anti-terrorism and in fugitive tracking. Therefore, we believe that this research is of great significance to the long-term interests of society.

To achieve a reliable identification of people in public spaces, it is necessary to overcome the problem of variability in pedestrian behaviors in such environments through the collection and identification of pedestrian gait information from multiple views [5,6]. Formal gait recognition uses 90° gait features that provide the most salient and comprehensive details of human posture as experimental data. The rationale for this method is that the gait characteristics in other views overlap due to the perspective problems of human physical characteristics, with the result that the contour characteristics are not effectively rendered. This is also one of the complications of multi-view gait recognition. Moreover, in practical terms, in order to preserve the advantages of passive identification, it is crucial not to establish a fixed walking position and camera viewpoint for the pedestrian [7]. This problem needs to be solved urgently.

In the task of multi-view gait recognition, when the angle of view moves from 90° to 0° and 180°, the contour of the human body is affected by the shooting angle, and

some of the gait feature information is lost. This significantly impacts the extraction of the gait contour characteristics. In response to this issue, this paper uses a Siamese neural network to calculate the posture relationship between the two views and calculate the characteristic conversion factor. Under the premise of retaining identity information, the useful high-dimensional intensive characteristics of the network are strengthened to make its high-dimensional features clearer, and the effect of the loss of gait characteristics on recognition accuracy is lessened. The SMViT model constructed using this concept can obtain higher recognition accuracy in a non-90° multi-view; moreover, this model is more robust.

In summary, this paper makes the following contributions:

(1) It designs a reasonable and novel gait-view conversion method, which can deal with the problem of multi-view gait;
(2) It constructs the SMViT model, and uses the view characteristic relation module to calculate the association between multi-view gait characteristics;
(3) We develop a gradually moving view-training strategy that can raise the model's robustness while raising the recognition rate for less precise gait-view data.

The structure of this paper is as follows. The technologies related to gait recognition are introduced in Section 2. Then, the SMViT is constructed and the gradual-moving-view training method is explained in Section 3. In Section 4, experimentation with the CASIA B gait dataset [8] is employed to explore the models and methods that are presented in this paper. Finally, in Section 5, we summarize the research contained herein and consider the future directions of gait recognition technology.

## 2. Related Work

At present, there are many methods of solving the problem of multi-view gait recognition. Some researchers adopt the method of constructing a three-dimensional model and use the close cooperation of multiple cameras to construct a three-dimensional model of pedestrian movement, so as to weaken the influence of multiple perspectives, clothing, and other factors. Bodor et al. proposed combining arbitrary views taken by multiple cameras to construct appropriate images that match the training view for pedestrian gait recognition [9]. Ariyanto et al. constructed a correlation energy map between their proposed generative gait model and the data, and adopted a dynamic programming method to select possible paths and extract gait kinematic trajectory features, proposing that the extracted features were inherent to the 3D data [10]. In addition, Tome et al. set out a comprehensive approach that combines the probability information related to 3D human poses with convolutional neural networks (CNNs), and introduced a unified formula to address the challenge of estimating 3D human poses from a single RGB image [11]. In subsequent research, Weng et al. changed the extraction method of human 3D pose modeling, and proposed a deformable pose ergodic convolution to optimize the convolution kernel of each joint by considering context joints with different weights [12]. However, this method of 3D pose modeling is more complicated to calculate and has high requirements regarding the number of cameras and the shooting environment, so it is difficult to use in application settings with ordinary cameras.

Some scholars used the view transformation model (VTM) to extract the frequency domain features of gait contours by transforming them from different views. For instance, Makihara et al. proposed a gait recognition method based on frequency domain features and view transformation. First, a spatio-temporal contour set of gait characteristics was constructed, and the periodic characteristics of gait were subjected to the Fourier analysis to extract the frequency domain features of pedestrian gait; the multi-view training set was used to calculate the view transformation model [13]. In this method, the Spatio-temporal gait images in the gait cycle are usually first fused into a gait energy image (GEI), which is a Spatio-temporal gait representation method first proposed by Han et al. [14]. Kusakunniran et al. combined the gait energy image (GEI) with the view transition model (VTM), and used a linear discriminant analysis (LDA) to optimize the feature vectors and

improve the performance of VTM [15]. Later, Kusakunniran et al. used a motion clustering method to classify gaits from different views into groups according to their correlation; within each group, a canonical correlation analysis (CCA) was used to further enhance the linear correlation between gaits from different views [16]. In addition, researchers have considered how to perform gait recognition from any view. Hu et al. proposed a viewpoint invariant discriminant projection (ViDP) method to improve the discrimination accuracy of gait features using linear projection [17]. However, most of these methods are realized by domain transformation or singular value decomposition, and the perspective of transformation is complicated.

Others have used an adversarial generative network to normalize multiple views into a common perspective. Zhang et al. proposed a perspective-shifting adversarial generative network (VT-GAN), which can transform gait views across two arbitrary views with only one model [18]. Shi et al. designed GaitGANv1 and GaitGANv2, versions of a gait adversarial generation network, which use GAN as a regressor to generate a standardized side view of a normal gait; this not only prevents the falsification of gait images, but also helps to maintain identity information, and the networks achieved good results in cross-view gait recognition [19,20]. In addition, Wen et al. used GAN to convert gait images with arbitrary decorations and views into normal states of 54°, 90°, and 126°, so as to extract view-invariant features and reduce the loss of feature information caused by view transformation [21]. Focusing on the problem of limited recognition accuracy arising from the lack of gait samples from different views, Chen et al. proposed a multi-view gait generation ad hoc network (MvGGAN) to generate false gait samples to expand the dataset and improve recognition accuracy [22]. However, the ability of this adversarial generative network structure to accurately undertake recognition tasks from the same perspective is easily affected by decorative features, such as clothes and backpacks, resulting in limited recognition accuracy.

## 3. SMViT and the Gradually Moving-View Training Method

### 3.1. Model Structure

In order to solve the problem of multi-perspective situations, this paper uses the Siamese neural network as a design basis and calculates the correlation between the characteristics of different views and uses this as the basis for the conversion of the characteristics of the view. When there are few specimens, a Siamese neural network can extract and learn the links between two groups of photos [23]. ViT is advantageous for the extraction of multi-scale features because of its robust strength and resistance to interference from mistaken samples [24,25]. In this paper, a two-channel Siamese module (Conv and MViT, CM Block) of convolution is constructed to extract the characteristics of multi-view gait contour features. The specific model structure is shown in Figure 1.

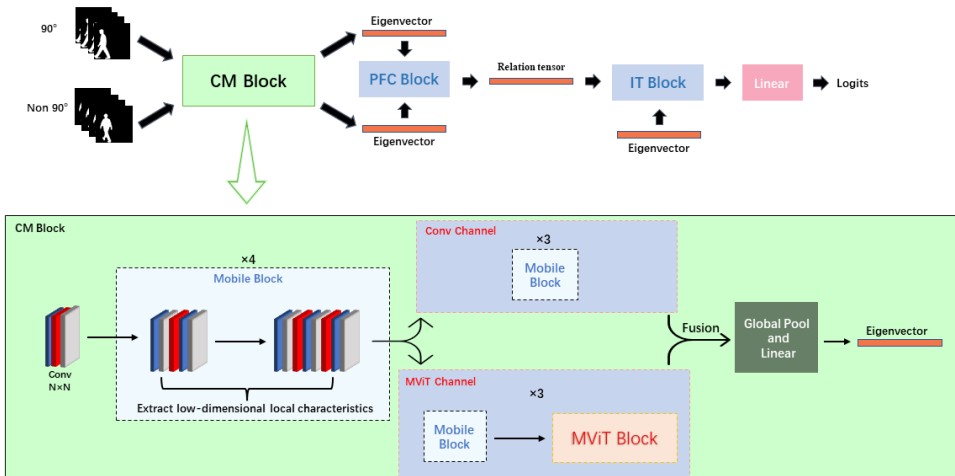

**Figure 1.** Structure diagram of SMViT.

In order to extract the gait information from various viewpoints, two feature extraction networks are used in the Siamese network module described in this paper. Convolution channels are used inside each module to obtain the contour's high-dimensional local features. Furthermore, we utilize the Mobile ViT channel to create high-dimensional states that are indicative long-distance attention characteristics of the current view.

In addition, the mobile view transformation (MVT) module is used to extract the view characteristics and tensors, meaning that the advantages of convolution and ViT are retained in the extraction of the gait contour features. This module is based on the Mobile ViT model, and incorporates the convolution with the transformer. Local processing is replaced with deeper global treatment in the convolution. In an effective receiving domain, we model long-distance non-local dependencies. The model has smaller parameters and produces ideal experimental results. The specific details of the module are shown in Figure 2.

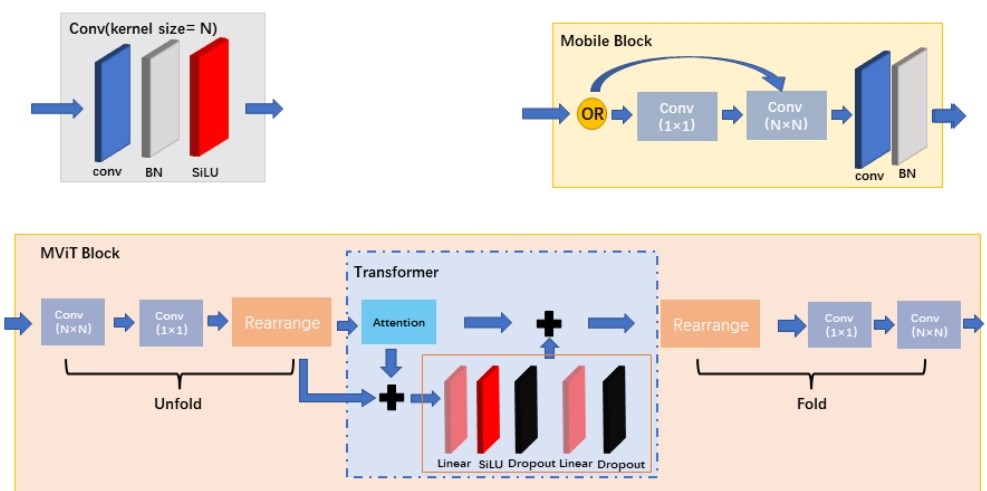

**Figure 2.** Details of the module.

As shown in Figure 2, there are two types of the conv block module (Conv Block), $N \times N$ convolution and point-by-point convolution, according to the difference in the convolution kernel size. This module consists of a convolution layer and a batch normalization layer. Mobile Block takes MobileNet as the essential conceptual basis [26–28] and controls the network's depth and the number of parameters by constructing depthseparable convolution. In addition, due to the differences in the processing methods and content of the feature extraction between the convolution and the ViT, the format conversion needs to be carried out before and after the transformer module in order to control the data processing format. *SiLU* is used as the activation function in each module, as shown in Equation (1), and the global average pooling layer is used in the pooling layer, as shown in Equation (2), where $x$ means the input matrix and $x_w$ represents the operation area of the pooling layer:

$$SiLU(x) = x \cdot Sigmoid(x) \tag{1}$$

$$Pooling(x_w) = Avg(x_w) \tag{2}$$

The transformer module absorbs some of the advantages of the convolutional computing and retains its characteristic processing capabilities in the space-perception domain. By dividing a large global receptive field into different patches in a non-overlapping way, $P = Wh$, where $w$ and $h$ are the width and height of the patches, respectively. Then, the transformer is used to encode the relationship between the patches. Specifically, the self-attention mechanism calculates the scaled dot-product attention by constructing the query vector $Q$, the value vector $V$, and the key vector $K$, as shown in Equations (3)–(5):

$$f(Q, K) = \frac{QK^T}{\sqrt{d_k}} \qquad (3)$$

$$X = softmax(f(Q, K)) \qquad (4)$$

$$Attention(X, V) = X \times V \qquad (5)$$

In this case, the module's computation cost of multi-head self-attention is $O(N^2 Pd)$. Compared with the traditional ViT, the calculation cost $O(N^2 d)$ is increased but its speed in practical applications is faster [29].

### 3.2. Perspective Feature Conversion Block and Inverse Transformation Block

The inverse transformation block (IT block) and the perspective feature conversion block (PFC block) are designed concurrently. The former is used to calculate the characteristics of the two perspectives obtained by the Siamese network, and the relation tensor is taken as the view conversion factor as shown in Formula (6). The latter is used to convert the high-dimensional characteristics between the two views as shown in Formula (7). Among them, *x* and *y* are two view-cornering characteristics and *N* is the capacity of the target view set. The process of calculating the gait characteristics from different perspectives in the PFC block and IT block are shown in Figure 3.

$$IT(x, y) = x + PFC(x, y) \qquad (6)$$

$$PFC(x, y) = \frac{\sum_{i=1}^{N}(x_i - y_i)}{N} \qquad (7)$$

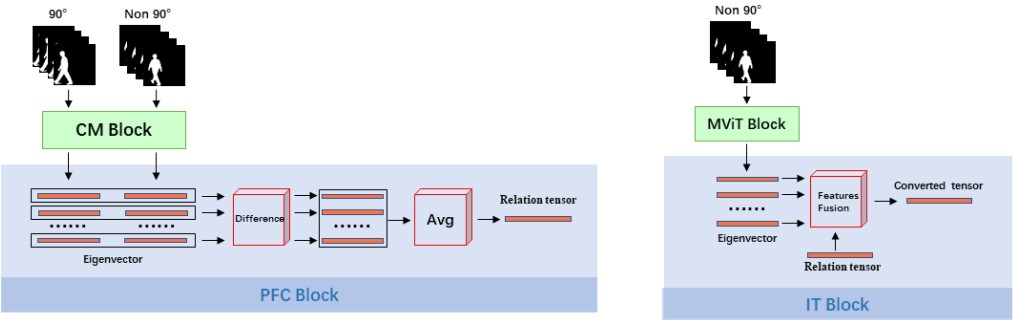

**Figure 3.** PFC block and IT block.

### 3.3. Gradually Moving-View Training Method

To develop the multi-view gait recognition model SMViT, this article designs a brand new, suitable, multi-view gait-recognition method; that is, the gradually moving-view training method. The training strategy of this method differs between the perspective feature relationship calculation module and the classification module.

In the characteristic view relationship calculation module, SMViT uses convolution and dual-channel VIT to extract the gait characteristics. To better calculate the difference between the 90°- and non-90°-perspective characteristics, the same pre-training weights are used to calculate the characteristics tensor of the perspective. Because the purpose of this module is to calculate the perspective characteristic tensor, not the classification, the pre-training weight of the last two layers of the network module needs to be eliminated. The specific training steps are shown in Algorithm 1.

---

**Algorithm 1:** Process of training the characteristic view relationship calculation module.

---

**Input:** CASIA B and CASIA C gait datasets.
**Step 1:** First, supplement the dual-channel view feature relationship calculation module as a complete classification model and conduct pre-training in the CASIA C dataset.
**Step 2:** Freeze the pre-training parameters to remove the weight of the final classification layer.
**Step 3:** Load the parameters from step 2 to the dual-channel perspective characteristic relationship calculation module.
**Step 4:** Use the module obtained in step 3 to extract the dual-channel features of 90° and non-90° CASIA B gait data.
**Step 5:** Calculate the relationship tensor between the two perspectives obtained in step 4 with the PFC block.
**Step 6:** Store the characteristic relationship between the two views obtained in step 5, and hand it over to the classification module.
**Output:** The characteristic relationship tensor between the two views.

---

In the classification module, the tensors of the gait characteristics of the different views after conversion should be identified and classified. Therefore, starting with the weight of the 90° model with high accuracy, training is undertaken in two directions of 0° and 180°. That is, a training weight of 90° is used as the initial weight of the model when training 72° and 108° gait data. When training at 54° and 126°, the training weights of, respectively, 72° and 108° are loaded and so on. This part uses a cross-entropy loss function as the method of loss calculation as shown in formula 8. The specific training steps are shown in Algorithm 2.

$$Loss(output, class) = weight_{[class]}\left(-output_{[class]} + \log\left(\sum_j e^{output_j}\right)\right) \tag{8}$$

Here, $output$ is the prediction result, $class$ is the actual label of this sample, and $output_{[class]}$ represents the element of the $class$ position in $output$, that is, the predicted value of the real classification. Finally, $weight_{[class]}$ is a weight parameter.

---

**Algorithm 2:** Classification Module Training Process

---

**Input:** CASIA B gait dataset.
**Step 1:** First, the 90° gait data are transformed and recognized (at this time, there is no change in the characteristics of the perspective), and the parameter weight is saved.
**Step 2:** The weight parameters obtained in step 1 are loaded to the classification module, the gait dataset (such as 72° and 108°) of the adjacent perspective is trained, and the parameter weight is saved.
**Step 3:** The characteristic relationship tensor between the two perspectives is matched and the parameter weights obtained in the previous step are loaded into the model.
**Step 4:** The trained perspective weight is loaded to the model, the gait dataset of adjacent non-90° perspectives is trained, and the weight parameters are saved.
**Step 5:** The classification layer and the regression layer are used to identify and classify the characteristic tensor of the view.
**Step 6:** Push in two directions (90°→0° and 90°→180°), and repeat steps 3, 4, and 5.
**Output:** The gait recognition model SMViT.

---

## 4. Experiment and Analysis

### 4.1. Experimental Data

The CASIA B dataset is a large dataset that is widely used in multi-view gait recognition tasks. It consists of 124 subjects (31 women and 93 men) [19]. The gait images of each subject in the three different states of normal walking (NM), walking with a bag (BG), and walking with a coat (CL) were collected [30] from 11 points of view, from 0° to 180° (with an interval of 18° for close views) as shown in Figure 4.

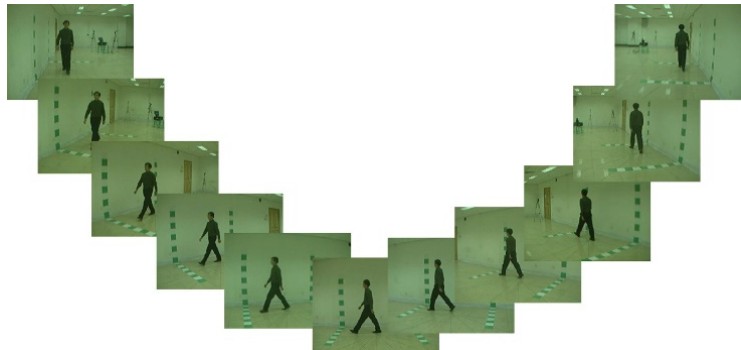

**Figure 4.** CASIA B multi-view gait dataset [8].

*4.2. Experimental Design*

In multi-view gait recognition tasks, due to the offset of the perspective, some human gait contour characteristics are lost and the recognition accuracy is reduced. This experiment designated 90° as a high-precision standard perspective. The remaining 10 views of the gait characteristics are calculated using the standard view relationship. The mutual verification between the perspectives is not considered; only the gait recognition accuracy inside the perspective is calculated. Additionally, a simple perspective conversion factor group is established to transform the view feature tensor with less feature information into a 90° feature tensor with more useful and distinct feature information.

In order to compare the results with the comparison model for the same data, we directly used the gait contour data provided by the CASIA B gait database. In addition, this allowed us to be more attuned to the uncertainty caused by people's attire and walking speeds, and other aspects of practical application scenarios. From the same perspective, we ignore slight differences in dress, walking speeds, and other personal features, and divide the overall data into the training set and the verification set according to the 7:3 ratio. There is no crossover between each view, in order to improve the recognition accuracy within each view. The gait data obtained in the actual application scenario may not necessarily contain a complete gait cycle, and the gait characteristics are random. Therefore, this experiment does not adopt the gait-cycle group as the input data. Instead, the gait group with three walking states is scattered at will to ensure that the model's effect is similar to a complex real-world environment.

Setting the initial learning rate to $1 \times 10^{-3}$, with Adam as the optimizer, we used the categorical_ The crossentropy multiclass cross-entropy loss function to calculate loss. In this experiment, Pycharm, an efficient Python IDE, was used to write code. The code was tested in Pytorch 1.8 and CUDA 11. The various equipment parameters used in the calculation process are shown in Table 1.

**Table 1.** Experimental environment.

| Environment | Parameter/Version |
|---|---|
| CPU | *I7-10700K* |
| GPU | NVIDIA RTX 3060 |
| CUDA | 11.0 |
| Pytorch | 1.8 |
| Operating System | Win10 |

*4.3. Experimental Results for the CASIA B Dataset*

To evaluate the effectiveness of the SMViT model and the view movement method (SMViT_T) proposed in this paper, we used the first 10,000 training loss changes of the two intermediate views as an assessment of the convergence effect. It can be seen that, even in the middle of the view offset, the SMViT model proposed in this paper can still

effectively converge and stabilize under the general trend as shown by the blue line in Figures 5 and 6. After the gradually moving-perspective training, not only is the model's drop in loss significantly improved, but the unstable jumping phenomenon of losses is also suppressed to a certain extent as shown by the orange line in Figures 5 and 6.

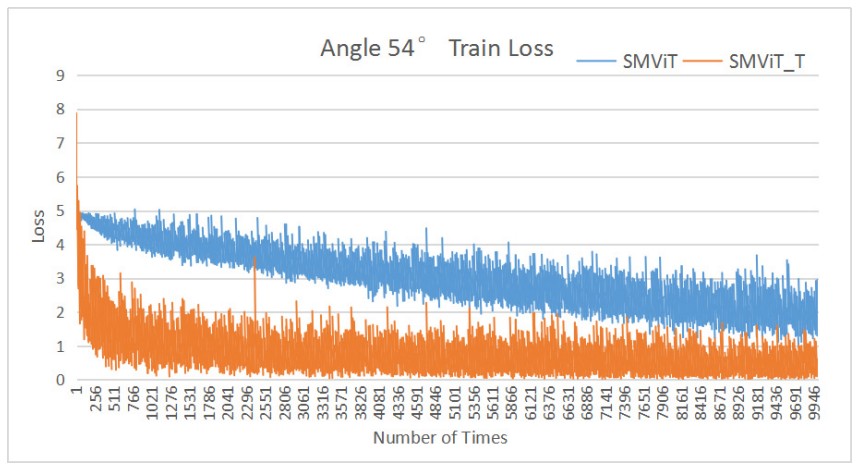

**Figure 5.** Loss change when the view is 54°.

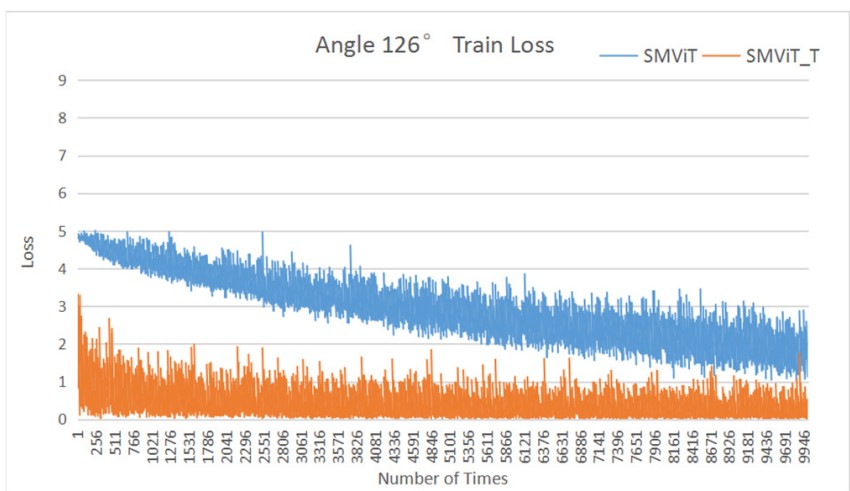

**Figure 6.** Loss change when the view is 126°.

The experiment was carried out at 10 angles, from 0 to 180 degrees, in order to prove that the model can effectively learn gait characteristics from multiple perspectives and overcome the problems caused by the poor learning effect and fluctuation in the accuracy rate, which are commonplace in multi-perspective gait recognition. Figure 7 shows the effect pictures of the model trained with or without gradually moving-view training for 10 views, but not for the 90° view. It can be observed that, except for the basic model (SMViT_BASE) at a perspective of 36°, there is a small oscillation in accuracy, and the experimental effects for the other views steadily increased. Both the base and T models quickly reach their peak accuracy and then stabilize as shown by the point line in Figure 7. The model proposed in this paper (SMViT_T), with gradually moving-view training, has a high accuracy for all the viewpoints, and there is no significant fluctuation in the recognition rate during the training process. The recognition rate of our model is always higher than that of the base model.

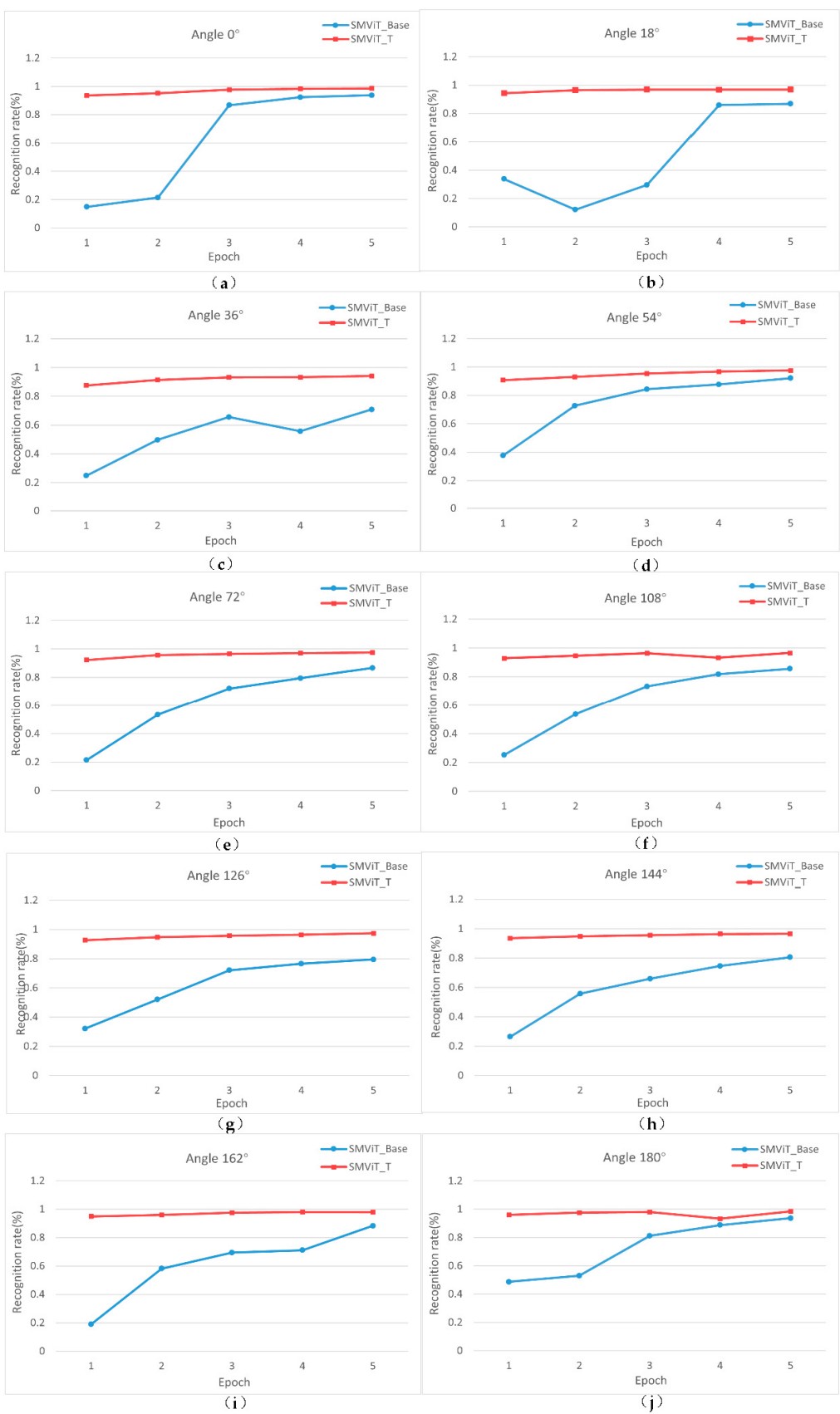

**Figure 7.** Diagram of the training process for the model proposed in this paper for various views in the CASIA B dataset (In the figure (**a**–**j**) are the experimental results of SMViT base and T in each angle).

### 4.4. Comparison with the Latest Technology

4.4.1. Ablation Experiment

At present, in many gait recognition studies, walking speed, clothing, and other characteristics are tested separately. As such, it can be considered an idealized experiment where some of the complexity of the data are eliminated by manual classification. This allows the model to obtain good results in NM classification. Other classifications, however, have poor accuracy. We suggest that, in real-world application scenarios, human appearance is highly uncertain. Therefore, the data obtained for different states are trained and verified separately, and the results cannot be used as the possible results of practical applications.

We experimented with a mixture of data for various characteristics to maximize the complexity of the gait data. In addition, our structural design focuses on improving the accuracy of gait recognition from multiple perspectives, rather than using cross-perspective experimental methods.

For the 11 views in the CASIA B dataset, our model (SMViT_Base) was compared with SPAE [31], GaitGAN [19,20], Multi_View GAN [21], Slack Allocation GAN [32], GAN based on U-Net [33], and PoseGait [34] in terms of the internal recognition rate of non-cross-view offset views. That is, without considering cross-verification, verification experiments only considered the multi-view perspective. Due to the limitations of the experimental environment and equipment, we could not effectively restore the experimental results of the multiple comparative models. Therefore, we directly used the experimental results presented in the papers. Other model data, shown in Table 2, were taken from the average value of the three-state gait recognition rates from the same view in the same dataset. It can be seen that, for all the views, the model presented here showed significant improvements when compared to the other gait recognition models. Additionally, the average upgrade index exceeded 20 percentage points. It is verified that, in the task of multi-view recognition with a non-crossing view, the model proposed in this paper is better than the selected comparison model.

**Table 2.** Precision comparison of CASIA B with the latest technology for each view.

| Comparison of Model Accuracy for Each View When Not Crossing Views | | | | | | | | | | | |
|---|---|---|---|---|---|---|---|---|---|---|---|
| | 0° | 18° | 36° | 54° | 72° | 90° | 108° | 126° | 144° | 162° | 180° |
| SPAE [31] | 0.7419 | 0.7661 | 0.7150 | 0.6989 | 0.7311 | 0.6801 | 0.6854 | 0.7258 | 0.7016 | 0.6881 | 0.7231 |
| GaitGANv1 [19] | 0.6828 | 0.7123 | 0.7285 | 0.7339 | 0.6962 | 0.7043 | 0.7150 | 0.7285 | 0.7204 | 0.7042 | 0.6828 |
| GaitGANv2 [20] | 0.7258 | 0.7554 | 0.7150 | 0.7332 | 0.7527 | 0.707 | 0.6962 | 0.7392 | 0.7150 | 0.7311 | 0.6989 |
| Multi_View GAN [21] | 0.7213 | 0.7869 | 0.7814 | 0.7589 | 0.7568 | 0.7131 | 0.7322 | 0.7431 | 0.7431 | 0.7480 | 0.7513 |
| Slack Allocation GAN [32] | 0.7473 | 0.7258 | 0.7258 | 0.7141 | 0.7560 | 0.7336 | 0.6967 | 0.7365 | 0.7277 | 0.7243 | 0.7221 |
| GAN based on U-Net [33] | 0.7365 | 0.7715 | 0.7956 | 0.7957 | 0.8521 | 0.7822 | 0.8172 | 0.7956 | 0.7984 | 0.7419 | 0.7580 |
| PoseGait [34] | 0.7231 | 0.7365 | 0.7688 | 0.7822 | 0.7446 | 0.7473 | 0.7607 | 0.7284 | 0.7553 | 0.7365 | 0.6586 |
| **SMViT_Base** | **0.9802** | **0.9704** | **0.9318** | **0.9805** | **0.9689** | **0.9744** | **0.9668** | **0.9617** | **0.9529** | **0.9451** | **0.9831** |

At the same time, the average values for the 11 views of normal walking (NM), walking with a backpack (BG), and walking while wearing a jacket (CL) were compared. For this comparison, the training sets and verification sets of each model were taken from the same view. The proportion of internal training sets and verification sets for each view is 7:3, and the cross-verification of the view is not considered. From Figure 8, it can be seen that the red model, which was proposed in this paper, significantly increased the average value of multi-view mixed recognition rates, which increased by about 20 percentage points.

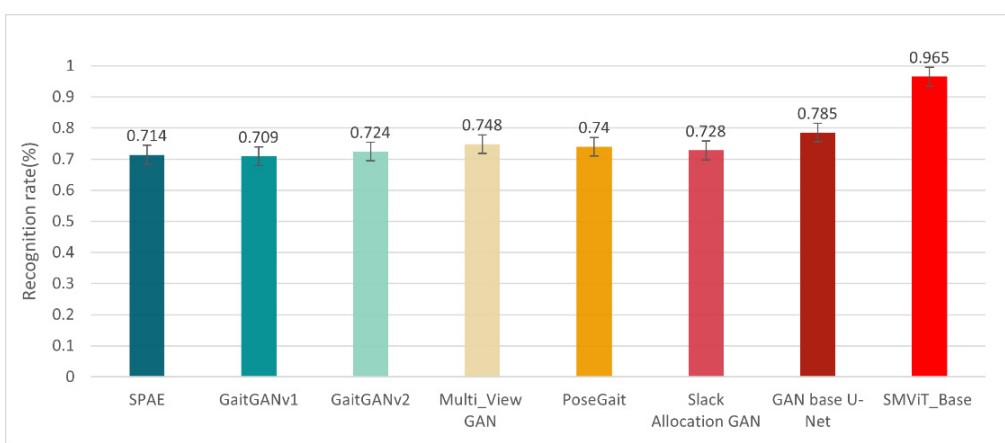

**Figure 8.** Comparison of the average validation rates of the model from multiple views.

### 4.4.2. Validation of the Gradually Moving-View Training Strategy

As shown in Figure 7, we initially demonstrated the effectiveness of this strategy by comparing the T model with the base model in SMViT. During the experiment, we found that, with this training strategy, SMViT can still maximize the stability of learning for some perspectives with low-quality data. Taking the 18° gait data as an example, we conducted ablation experiments to verify the effectiveness of the gradually moving-view training method. As shown by the square point line in Figure 9, during the first 15 rounds of training, the SMViT_base model proposed in this paper dropped significantly in the second and ninth rounds and there was a significant saturation of recognition rates. After the experimental analysis, we surmised that the first decline was due to the loss of a number of gait characteristic outlines. At the same time, this model does not use view-mobilization training methods to convert and strengthen the gait characteristics, making the model unable to effectively learn the characteristics, and the accuracy decreases sharply. The second drop is due to the small number of features, which led to the abnormal situation of gradual overfitting in the verification accuracy; this also reflects the improvement in the model's robustness facilitated by the view-transition training method. At around 13 rounds, the base model reaches the upper limit of saturation accuracy but is still about one percentage point lower than the SMViT_T model's recognition accuracy. On the whole, due to the gradually moving-view training strategy of SMViT, the initial recognition rate is about 70 percentage points higher than that of the basic model; our model also maintains a relatively stable level of recognition accuracy. Although the accuracy saturation trend also appeared quickly, the upper limit of the saturation value was about one percentage point higher than that of the base model, and the oscillation amplitude of the validation rate remained below one percentage point.

In this paper, we integrated the design concept of Siamese neural networks and a variant mobile vision transformer model and built a multi-view Siamese ViT gait recognition model: SMViT. At the same time, we designed a gradually moving-view training strategy for multi-view gait recognition, referred to as SMViT_Base and SMViT_T. After conducting a number of experiments on the CASIA B dataset, it was shown that the Siamese feature relationship calculation method can be used to obtain the perspective characteristic conversion factor, which can be used to determine the relationship between different perspective gait characteristics; this effectively improves the accuracy of multi-perspective gait recognition. Our experimental results show that the proposed model can significantly improve the recognition rate when compared with the existing generative multi-view gait recognition methods, without considering cross-view verification. We demonstrated an increase of 20 percentage points in the hybrid recognition rate, without considering the external attire of the pedestrians. Therefore, SMViT expands the gait recognition view while ensuring high accuracy, improving efficient gait recognition in multi-view practical application scenarios.

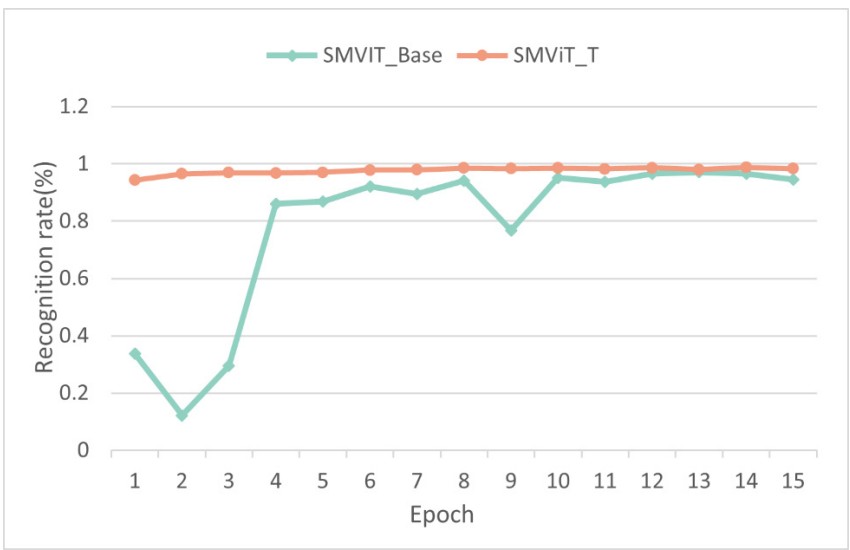

**Figure 9.** Reliability verification of the gradually moving-view training method at 18°.

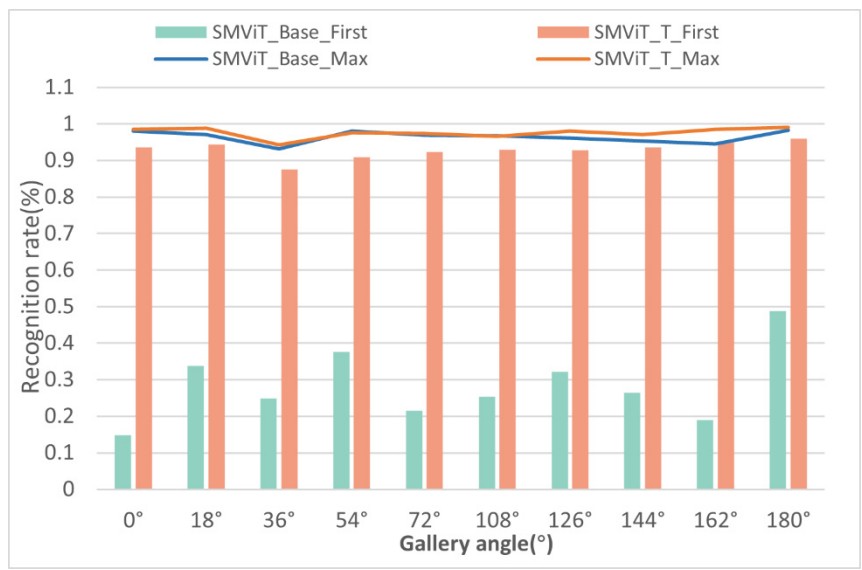

**Figure 10.** Comparison of the initial validation rate and the maximum validation rate of the proposed model trained with or without gradually moving-view training.

## 5. Conclusions and Future Prospects

In the future, a more abundant dataset can be used to verify the recognition effect, and a more sophisticated view-feature conversion module can be used to enhance the application scope of SMViT. Additionally, when the visible light intensity is insufficient, the infrared thermal imaging target tracking method can be used to extract the gait contour features, creating the possibility of dealing with more complex and variable natural environments [35] and undertaking the tracking of more obscure gaits [36,37]. We believe that the design of SMViT with multiple covariates will open up new methods for multi-view gait recognition; the vision transformer can also play a role in multi-view gait recognition tasks in complex environments.

**Author Contributions:** Data curation, L.Y.; investigation, R.L.; methodology, Y.Y. and L.Y.; resources, L.Y.; supervision, L.Y. and K.W.; validation, Y.Y. and R.L.; writing—review and editing, Y.Y., L.Y. and F.C. All authors have read and agreed to the published version of the manuscript.

**Funding:** This research was funded by the Key Projects of Yunnan Applied Basic Research Plan, grant number 2018FA033.

**Institutional Review Board Statement:** Not applicable.

**Informed Consent Statement:** Not applicable.

**Data Availability Statement:** The datasets analyzed in the current study are available in the CASIA gait database of the Chinese Academy of Sciences: http://www.cbsr.ia.ac.cn/china/Gait%20 Databases%20CH.asp (Website viewed on 16 October 2021). The data used in the experiment described in this article come from the CASIA gait database provided by the Institute of Automation of the Chinese Academy of Sciences.

**Conflicts of Interest:** The authors declare no conflict of interest.

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
