# Peer review of "Multi-View Gait Recognition Based on a Siamese Vision Transformer"

_applsci, doi:10.3390/app13042273_

Round 1

Reviewer 1 Report

The paper deals with multi-view gait recognition, proposing a novel approach based on Siamese neural networks and Vision Transfomers. In particular, it investigates how the relationships computed between two different views through a Siamese neural network can be used to improve recognition performance and robustness.

The paper deals with a topic that has been stimulating research interest in recent years, and has a strong impact on computer vision.

The literature review is coherent with the problems and the enhancements that the authors propose in the paper. 

Regarding clarity, the paper shows several issues. The text is not always clear to read and understand, with many sentences not well connected or left unfinished. Just to mention a few examples:

- line 30 "resolution instruments without the perception of others", it is not clear from the sentence who or what "others" are referring to;

- line 45-47 "when the view is shifted from 90° to 0° and 180°, it will be gone with the view so that when the outline of the walking posture is extracted, the degree of coincidence of the body part is U-shaped", the second part of the sentence is very hard to understand;

- line 145-146 "Not losing the order of the patch nor affecting the inner space relation of the pixels in each patch.", this sentence seems incomplete; it should be better connected to the rest;

- line 148-149 "Come to retain the advantages of convolution and ViT in the extraction of gait contour features.", another sentence that appear not well connected (and meaningless so isolated);

- lines 292-296 "Taking the 18° gait data ... of the view gradually moving training, As can be seen from"

The lack of clarity in the writing is the paper's biggest problem, making it very difficult to understand the motivation behind the work and the main contributions proposed. For example, the importance of the research topic is not well explained in the introduction, but only vaguely hinted at:

- line 33 "and has an amazing advantage", is a too vague sentence, what is that "amazing" advantage? it should be better described and discussed;

-line 34 "gait recognition is of great significance to the long-term steadiness of national society", what does it mean?

In the methods section (Section 3), the various modules are presented one after the other without a clear connection between all the various parts. For example, section 3.1 and Figure 1 present an overview of the proposed approach with Siamese and Transfomer networks, but no mention is made of the PFC and IT modules; such modules first appear in section 3.2 after a lengthy description of the transformer, and it is difficult to see how it ties in with the rest of the proposed system.

In section 3.2, the IT and PFC blocks are introduced, but their description is somewhat confusing. In line 184 the authors say that "the former" (thus the IT block) is described by formula (6), which, however, refers to the PFC block; the same for PFC, described by reference to formula (7), which, however, defines the IT block.

In addition, it can be seen from Figure 1 that the input to the system are sequences of silhouettes; how these silhouettes are obtained is not described in the paper.

The experiments section also presents several problems. The authors evaluate their approach on the public CASIA-B dataset, but in section 4.2 they describe a division of the dataset into training and test sets that does not appear to be the official division of the CASIA-B dataset. The actual division chosen by the authors is then not very clear: they indicate a 7:3 division (i.e., 70% of the data for training?), but it is not clear whether applied by grouping different subjects or different views.

In section 4.4.1 the proposed method is compared with the state of the art. Again, it is not clear how this comparison was made, since the methods mentioned are generally evaluated differently. For example, in their respective papers SPAE and GaitGAN present their results through three tables (normal walking, walking with bag, walking wearing a coat) that include a combination of "gallery set view" and "probe set view".

How were the values provided in Table 2 for each state-of-the-art method obtained? Were they calculated from scratch on the division of data proposed by the authors in section 4.2? Or are they values reported for of each method extracted from the corresponding paper?

If so, please explain how this computation was done, since for methods such as GaitGAN it is not clear how the values shown in Table 2 are related to the values presented in the corresponding paper [19].

The results reported in Figure 5 show the recognition accuracy of the base and "moving training" models for different view angles. However, the results provided refer to the training process and it is not clear what information these results should provide. At first glance it seems to say that the developed network overfitted easily on the training set, while it would have been more interesting to see the accuracy results on the test set for each corner. Please provide more details and explanations of that experiment so that we can better appreciate the reported results. 

Finally, nowhere in Section 4 regarding experiments are the implementation details of the proposed method described. What kind of deep learning framework was used (e.g. Keras, Pytorch)? What hyperparameters were used to train the networks? What kind of hardware was used?

All these implementation details are important information to provide to make the proposed experiments replicable by other researchers in the field.

Minor comments:

- line 50, possibile repetions in "two views of the two views and calculate"

- line 126, the title of subsection 3.1 is "Subsection"; please choose a more meaningful title

- line 196, the model proposed is named SMVIT instead of SMViT. Please correct the paper so that the same acronym is used throughout.

- line 229, the figure 4 is taken from the CASIA-B paper. It should be acknowledged using a reference in the caption.

- Table 2, all the reference for the state-of-the-art methods considered are wrong (e.g., SPAE [23] should be SPAE [31])

- Usually "ablation study" refers to a set of experiments to further investigate network modifications or the role of some hyper-parameters, but in this case "ablation experiment" is used as the title of the section containing the comparison with the state-of-the-art; I would suggest to move section 4.4.1 before section 4.4.

Author Response

Dear reviewers:

Thanks very much for your time to review this manuscript. I really appreciate all yourcomments and suggestions. We have considered these comments carefully and triedour best to address every one of them.

- line 30 "resolution instruments without the perception of others", it is not clear from the sentence who or what "others" are referring to;

Our response: We thank the reviewer for the very interesting comment. In fact, here is a description of the advantages of gait recognition in reference [2]. "Others" here can be understood as the identified target. To avoid ambiguity, we replace "others" with "identified target".

- line 45-47 "when the view is shifted from 90° to 0° and 180°, it will be gone with the view so that when the outline of the walking posture is extracted, the degree of coincidence of the body part is U-shaped", the second part of the sentence is very hard to understand;

Our response: Here, we actually want to express that "when the angle of view moves from 90 °to 0° and 180°, the contour of the human body will be affected by the shooting angle, and part of the gait feature information will be lost.",It is intended to use “U-shaped” to represent the change curve of the richness of gait feature information.

- line 145-146 "Not losing the order of the patch nor affecting the inner space relation of the pixels in each patch.", this sentence seems incomplete; it should be better connected to the rest;

- line 148-149 "Come to retain the advantages of convolution and ViT in the extraction of gait contour features.", another sentence that appear not well connected (and meaningless so isolated);

Our response: We deleted the sentences that may cause ambiguity and adjusted the sentence structure. The meaning of this paragraph is that the gait contour extraction module combines the feature extraction advantages of convolution and ViT, and explains the role of the processing module.

- lines 292-296 "Taking the 18° gait data ... of the view gradually moving training, As can be seen from"

Our response: The meaning here is that in the process of verifying the effectiveness of the angle shift training, the accuracy of many angles is unstable and jumps. We take the 18 ° experimental results as an example to show the effectiveness of the training strategy.

- line 33 "and has an amazing advantage", is a too vague sentence, what is that "amazing" advantage? it should be better described and discussed;

Our response: "Amazing" here means that gait recognition can track the itinerary of fugitives without disturbing the suspect. This is an advantage that other biometric recognition technologies do not have. Therefore, many researchers including us believe that gait recognition is a technology with infinite potential. We want to use "amazing" to express this view.

-line 34 "gait recognition is of great significance to the long-term steadiness of national society", what does it mean?

Our response: As stated above, gait recognition research can make great effect in tracking fugitives and anti-terrorism. Therefore, we believe that the research of gait recognition is of great significance for maintaining the stability of the country and society.

In the methods section (Section 3), the various modules are presented one after the other without a clear connection between all the various parts. For example, section 3.1 and Figure 1 present an overview of the proposed approach with Siamese and Transfomer networks, but no mention is made of the PFC and IT modules; such modules first appear in section 3.2 after a lengthy description of the transformer, and it is difficult to see how it ties in with the rest of the proposed system.

Our response: According to your suggestions on writing ideas, we would like to explain as follows:Firstly, we used to introduce the overall structure of SMViT, and then explain the key modules.Firstly, the model structure proposed in this paper is described in a framework, and then the main technical points of the framework are explained.After that, we use the formula to explain the IT module and PFC module. As the key nodes connecting the feature extraction module and the discrimination module in SMViT, they are responsible for the conversion and utilization of the feature tensor of the perspective. Therefore, we believe that the two modules should be introduced separately after the model structure description. In addition, we describe the functions and positions of the two modules in SMViT in the first paragraph of 3.2.

In section 3.2, the IT and PFC blocks are introduced, but their description is somewhat confusing. In line 184 the authors say that "the former" (thus the IT block) is described by formula (6), which, however, refers to the PFC block; the same for PFC, described by reference to formula (7), which, however, defines the IT block.

Our response: Thank you for your correction. We have corrected this in the article.

In addition, it can be seen from Figure 1 that the input to the system are sequences of silhouettes; how these silhouettes are obtained is not described in the paper.

Our response:The picture here is provided on the official website of the Institute of Automation of the Chinese Academy of Sciences. Since the following comparison models use the same gait contour map, in order to control the variables, we directly use this part of data.

The experiments section also presents several problems. The authors evaluate their approach on the public CASIA-B dataset, but in section 4.2 they describe a division of the dataset into training and test sets that does not appear to be the official division of the CASIA-B dataset. The actual division chosen by the authors is then not very clear: they indicate a 7:3 division (i.e., 70% of the data for training?), but it is not clear whether applied by grouping different subjects or different views.

Our response: Due to the limitation of gait recognition data and experimental environment, CASIA B data set is selected as the experimental data. It is very common to segment a single dataset as model training data and validation data. And it will not affect the validity and correctness of the results. In addition, in order to be more close to the uncertainty of people's dressing, walking speed and other aspects in practical application scenarios. In the same perspective, we ignore the slight differences in dressing, walking speed and other aspects of different people, and divide the overall data into training set and verification set according to the 7:3 ratio.

In section 4.4.1 the proposed method is compared with the state of the art. Again, it is not clear how this comparison was made, since the methods mentioned are generally evaluated differently. For example, in their respective papers SPAE and GaitGAN present their results through three tables (normal walking, walking with bag, walking wearing a coat) that include a combination of "gallery set view" and "probe set view".

How were the values provided in Table 2 for each state-of-the-art method obtained? Were they calculated from scratch on the division of data proposed by the authors in section 4.2? Or are they values reported for of each method extracted from the corresponding paper?

If so, please explain how this computation was done, since for methods such as GaitGAN it is not clear how the values shown in Table 2 are related to the values presented in the corresponding paper [19].

Our response: We believe that in real application scenarios, human appearance is highly uncertain. Therefore, the data in different states are trained and verified separately, and the results can not be used as the possible results of practical applications. However, due to the limitation of experimental equipment, it is difficult for us to reproduce many models. Therefore, We directly used the result data in these comparative model papers, and take the average of the results of various environments in these papers as the accurate value in the case of mixed data (this accurate value is often higher than the actual verification accurate value). In addition, our research focuses on improving the accuracy of gait recognition from multiple perspectives. Therefore, we only compare these models within the same perspective. 

The results reported in Figure 5 show the recognition accuracy of the base and "moving training" models for different view angles. However, the results provided refer to the training process and it is not clear what information these results should provide. At first glance it seems to say that the developed network overfitted easily on the training set, while it would have been more interesting to see the accuracy results on the test set for each corner. Please provide more details and explanations of that experiment so that we can better appreciate the reported results.

Our response: As mentioned in the article, it means to verify the effect of the training strategy of gradual shift of view in multiple different angles to prove that this training strategy is an available and effective method for multi-view gait recognition. It can effectively solve the problems of slow learning speed and instability in multi-angle gait recognition tasks. By showing the fitting process of multi-angle gait features, you can more intuitively see the effect of this training method.

- line 50, possibile repetions in "two views of the two views and calculate"

Our response: Thank you for your correction. We have corrected this in the article.

- line 126, the title of subsection 3.1 is "Subsection"; please choose a more meaningful title

Our response: Thank you for your correction. We change "Subsection" to "model structure"in the article.

- line 196, the model proposed is named SMVIT instead of SMViT. Please correct the paper so that the same acronym is used throughout.

Our response: Thank you for your correction. We have corrected this in the article.

- line 229, the figure 4 is taken from the CASIA-B paper. It should be acknowledged using a reference in the caption.

Our response: Thank you for your correction. We have corrected this in the article.

- Table 2, all the reference for the state-of-the-art methods considered are wrong (e.g., SPAE [23] should be SPAE [31])

Our response: Thank you for your correction. We have corrected this in the article.

- Usually "ablation study" refers to a set of experiments to further investigate network modifications or the role of some hyper-parameters, but in this case "ablation experiment" is used as the title of the section containing the comparison with the state-of-the-art; I would suggest to move section 4.4.1 before section 4.4.

Our response: Thank you for your correction. We have corrected this in the article.

Reviewer 2 Report

The fundamental problem with this paper is that the description of the experiment is not detailed and precise enough for me.

The process of training and testing is not separated. I have a feeling that the same data set is involved in both training and testing. As a result of the testing process, we can see the comparising table of the precision of the algorithms published in the literature and in this paper.

Can these results be compared? Were the same benchmark used for testing? Precision is not an informative enough measure, what about sensitivity and accuracy? Or what does precision mean here?

There is no Table 1 in the paper, but there is a reference to it. I think the Table 2 is the Table 1. 

It would be useful to give an example when the algorithm does not work well and to point out what causes the error in the recognition.

Let us summarize my opinion. The Section 4 ought to be extended and corrected.

Author Response

Dear reviewers:

First of all, We thank you for the time and effort that they have put into reviewing the previous version of the manuscript. Their suggestions have enabled us to improve our work.We respond to your questions as follows:

The process of training and testing is not separated. I have a feeling that the same data set is involved in both training and testing. As a result of the testing process, we can see the comparising table of the precision of the algorithms published in the literature and in this paper.

Our response: Due to the limitation of gait recognition data and experimental environment, CASIA B data set is selected as the experimental data. It is very common to segment a single dataset as model training data and validation data. And it will not affect the validity and correctness of the results. This means that we train and verify under the CASIA B dataset, but through data segmentation, we ensure that the training and verification data are not the same. In addition, in order to be more close to the uncertainty of people's dressing, walking speed and other aspects in practical application scenarios. In the same perspective, we ignore the slight differences in dressing, walking speed and other aspects of different people, and divide the overall data into training set and verification set according to the 7:3 ratio.

Can these results be compared? Were the same benchmark used for testing? Precision is not an informative enough measure, what about sensitivity and accuracy? Or what does precision mean here?

Our response: We believe that in real application scenarios, human appearance is highly uncertain. Therefore, the data in different states are trained and verified separately, and the results can not be used as the possible results of practical applications. However, due to the limitation of experimental equipment, it is difficult for us to reproduce many models. Therefore,We use the same data as these comparative models to conduct the test. And directly used the result data in these comparative model papers, and take the average of the results of various environments in these papers as the accurate value in the case of mixed data (this accurate value is often higher than the actual verification accurate value). In addition, our research focuses on improving the accuracy of gait recognition from multiple perspectives.

There is no Table 1 in the paper, but there is a reference to it. I think the Table 2 is the Table 1. 

Our response: Thank you for your correction. We have corrected this in the article.

Round 2

Reviewer 1 Report

Authors resolved most of the comments left by reviewers. The readability and clarity of the paper have improved, although some sentences still remain unclear due to spelling and grammatical errors. For example:

- line 242 "with less feature information, To the 90° feature" there is a capital letter after a comma; it is not clear whether the capital letter signals the beginning of a new sentence or not

I suggest rereading the entire article to correct these errors and improve English clarity; consider possible use of online tools to help improve written English.

The main comments to the paper regarding the division of the dataset and experiments were considered only in the letter to the reviewers. This helps the reviewers better understand the authors' choices, but this information should also be reflected in the paper to make it clearer and more understandable to the research community.

Please report the explanations given to the reviewers in the paper as well.

One reviewer's comment was completely ignored: "Finally, nowhere in Section 4 regarding experiments are the implementation details of the proposed method described. What kind of deep learning framework was used (e.g. Keras, Pytorch)? What hyperparameters were used to train the networks? What kind of hardware was used? All these implementation details are important information to provide to make the proposed experiments replicable by other researchers in the field."

Please add details on that in the paper.

Author Response

Dear reviewers:

Thank you very much for your guidance in revising my paper in addition to your work and life. Your comments are very pertinent and helpful in enriching the details of my paper.In response to your question, I have made some modifications and adjustments to Chapter 4 of this article.It includes:

  1. Write out the details of the experimental structure design and the experimental environment;
  2. Add detailed explanations for some possible misconceptions, such as data set partitioning and interpretation of multi-angle accuracy comparison maps.

In addition, I have corrected your question in the article.Thank you again for your generous guidance and wish you success in your work.

Reviewer 2 Report

Basically, nothing has changed in the paper, only minor errors have been corrected.

I believe that the answers to the questions I asked should be reflected in the changes in the paper. In other words, short blocks should be included in the manuscript that draw the reader's attention so that these questions do not cross their mind.

I cancel my opinion about making significant changes to the fourth section, but the answers to my questions should appear in the manuscript as clarifications for certain parts.

Author Response

Dear reviewers:

Thank you very much for your guidance in revising my paper in addition to your work and life. Your comments are very pertinent and helpful in enriching the details of my paper.In response to your question, I have made some modifications and adjustments to Chapter 4 of this article.It includes:

  1. Write out the details of the experimental structure design and the experimental environment;
  2. Add detailed explanations for some possible misconceptions, such as data set partitioning and interpretation of multi-angle accuracy comparison maps.

These are revised and marked in the paper.Thank you again for your generous guidance and wish you success in your work.
